# Effects of the Face/Core Layer Ratio on the Mechanical Properties of 3D Printed Wood/Polylactic Acid (PLA) Green Biocomposite Panels with a Gyroid Core

**DOI:** 10.3390/polym12122929

**Published:** 2020-12-07

**Authors:** Nadir Ayrilmis, Rajini Nagarajan, Manja Kitek Kuzman

**Affiliations:** 1Department of Wood Mechanics and Technology, Forestry Faculty, Istanbul University-Cerrahpasa, Bahcekoy, Sariyer, Istanbul 34473, Turkey; 2Department of Mechanical Engineering, Kalasalingam Academy of Research and Education, Krishnankoil 626 126, Tamilnadu, India; n.rajini@klu.ac.in; 3Department of Wood Science and Technology, Biotechnical Faculty, University of Ljubljana, Jamnikarjeva 101, SI-1000 Ljubljana, Slovenia

**Keywords:** 3D printing, wood, polylactic acid, lightweight, mechanical properties, design, shell ratio, gyroid core

## Abstract

Gyroid structured green biocomposites with different thickness face layers (0.5, 1, 2 and 2.5 mm) were additively manufactured from wood/ polylactic acid (PLA) filaments using a 3D printer. The mechanical properties of the composite panels, bending properties, compressive strength (parallel to the surface), Brinell hardness, and face screw withdrawal resistance, were determined. The surface layer thickness significantly affects the mechanical properties of the composite materials. As the surface layer thickness was increased from 0.5 to 2.5 mm, all the mechanical properties significantly improved. In particular, the Brinell hardness and face screw withdrawal resistance of the specimens improved sharply when the skin thickness was higher than 2 mm. The bending strength, bending modulus, compressive strength (parallel to the surface), Brinell hardness, and face screw withdrawal resistance of the specimens with a skin of 0.5 mm were found to be 8.10, 847.5, 3.52, 2.12 and 445 N, respectively, while they were found to be 65.8, 11.82, 2492.2, 14.62, 26 and 1475 N for the specimens with a 2.5 mm skin. Based on the findings from the present study, gyroid structured composites with a thickness of 2 mm or higher are recommended due to their better mechanical properties as compared to the composites with skins that are thinner.

## 1. Introduction

Additive manufacturing, also known as 3D printing, refers to the process of producing 3-dimensional complex shaped solid objects from a computer file, such as with the Solidworks software. The use of 3D printing technology can provide numerous benefits such as fast production, single step production, low cost of labour, greater complexity, and more design freedom. The details of the 3D printing process are given in Figure 1 [1]. Traditional production methods, such as computer numerical control (CNC) milling or turning, create significant waste material from an initial block during the process [2,3,4]. However, additive manufacturing creates the final shape of the product by adding material instead of removing it. In this way, more efficient raw material use is ensured with a high level of geometric precision, while there is minimum material loss compared to traditional production methods. Other significant benefits of 3D printing technology are its low equipment cost and simple machine structure, and the wide variety of polymeric materials that can used [5,6,7]. Unlike traditional manufacturing techniques, such as machining or pressing part of the materials, with 3D printing complex shaped products are produced layer by layer from filaments [8,9].

Sandwich composites with a honeycomb core are widely used in the manufacture of composites in the construction, furniture, marine, sports, and aerospace industries. Due to high stiffness-to-weight and strength-to-weight ratios, the use of sandwich structure composites has increased considerably in recent years [10,11]. Although massive composites have excellent mechanical properties, their high mass, higher transport cost, abrasiveness with regard to processing tools, and high consumption of natural resources, make lightweight panels more attractive for many industries. In particular, many composite manufacturers now focus on the selection and design of skin and core materials. Shear strength is the most important strength for the core layer, while tensile and compressive strength are responsible for the load carrying capability of the composites. Moreover, the skin thickness and density, core density and structure all significantly affect the mechanical properties of the lightweight composites. 

One of the most used lightweight core structures for lightweight composites is the gyroid, which is a member of the triply periodic minimal surfaces (TPMS) family and as first identified by Alan Schonen in 1970 [12]. It presents the topology of an open celled foam [13]. Because of the gyroid’s complex structure it is only possible to produce it by additive manufacturing [14]. For example, Maskery et al. [15] investigated the energy absorption and various failure modes of a gyroid-structure prepared from Al-Si10-Mg. Moreover, Jung et al. [16] suggested the use of a gyroid-shaped core made from 3D porous graphene to obtain lightweight composites. One of the most interesting properties of the gyroid structure is its behaviour in compression [12].

The surface layer density and its thickness significantly affect the mechanical properties of composite materials, especially those of bending. Moreover, wood has a number of significant advantages when used as a reinforcing filler for thermoplastic composites, such as the low cost, easy supply, high modulus, problem free-disposal, and good machinability. Polylactic acid (PLA) is an environmentally friendly polymer and one of the most used filaments with fused deposition modelling (FDM) printers. The use of wood/PLA composites has increased considerably in the biocomposite industry in recent years, because of their significant advantages as compared to wood-filled petroleum-based polymer composites. Wood/PLA composites are mostly produced by extrusion, injection moulding, and hot-press moulding. As compared to the solid wood/PLA composites, lightweight wood/PLA composites are better for use when weight/strength is important. The skin/core layer ratio is one of the most important parameters affecting the mechanical properties of such composites. The ratio of the surface/core layer that can meet the mechanical properties required by the place of use must thus be determined, as otherwise an unsuitable surface/core layer ratio may result in failure under load. A comprehensive search of the literature revealed no studies on the mechanical properties of lightweight composites produced from wood/PLA. The current study thus focused on the impact of the shell ratio on the mechanical properties of the 3D printed lightweight wood/PLA composite panels with a gyroid core structure. The total thickness of top and bottom layers was gradually increased from 0.5 to 2.5 mm.

## 2. Materials and Methods

### 2.1. Manufacture of 3D Printed Wood-PLA Specimens 

The 3D printed test specimens with different face/core shell ratios were produced from commercial wood/PLA filaments with a diameter of 1.75 mm supplied by a local seller (Plastika Trček, Ljubljana, Slovenia). All of the specimens were printed on a desktop FDM 3D printer (Model: i3 MK3S, Prusa, Prague, Czech Republic), and designed within the accompanying Prusa Slicer software. The lattice structure, cell wall and cell length of the gyroid-shaped wood/PLA composite panels is shown in Figure 2. The specimens were manufactured without support in the base according to the FDM process. The experimental design is given in Table 1. 

The infill ratio in the specimens was kept constant at 15%. The printing temperature, resolution and nozzle diameter were 210 °C and 0.1 mm, and 0.4 mm, respectively. The gyroid infill pattern was selected in the core production of the specimens. 

Two specimens with dimensions of 100 mm × 100 mm × 10 mm (thickness) were manufactured for each type of treatment. The size of the gyroid structure is presented in Figure 3. The weight of all the specimen groups varied from 19.4 g to 65.8 g.

### 2.2. Determination of Mechanical Properties

The specimens were conditioned in a climate room (20 °C and 50% relative humidity) according to the ISO 291 [18] standard prior to testing. Bending strength and modulus, compressive strength (parallel to surface), face screw withdrawal resistance (SWR) and Brinell hardness were determined. The bending strength and bending modulus of the specimens with dimensions of 100 mm × 10 mm × 10 mm were determined with a crosshead speed of 5 mm/min according to the ISO 178 standard (span: 80 mm) [19]. The compressive strength (parallel to the surface) was examined on the specimens with dimensions of 50 mm × 10 mm × 10 mm with a crosshead speed of 2.5 mm/min according to the ASTM C364/C364M standard [20]. The Brinell hardness of the specimens with dimensions of 50 mm × 35 mm × 10 mm was determined in accordance with EN 1534 [21]. The load applied to the specimen surface by the indenter was 400 N. The surface SWR of the specimens with dimensions of 45 mm × 40 mm × 10 mm was determined using the ISO 27,528 [22] standard. The pilot hole driven into the specimens before the screw specified in the standard was hand-driven to 20 mm into each specimen. Two specimens were fixed to each other using glue before the screw test. The mechanical tests were carried out using a Lloyd Universal Testing Machine (Lloyd Instruments, West Sussex, UK). The numbers of test specimens for each treatment were as follows: four specimens were used to assess the bending strength/bending modulus, two for compressive strength (parallel to the surface), three for the SWR, and two for the Brinell hardness. 

### 2.3. Statistical Analysis

Test results were statistically evaluated using an analysis of variance at a 95% confidence interval. Significant differences in the mean values of the treatment groups were determined by Duncan’s multiple range test.

## 3. Results and Discussion

The mechanical properties of the specimens are given in Table 2. The bending strength and modulus of the 3D printed gyroid structured specimens without a face layer were found to be 1.76 N/mm^2^ and 324 N/mm^2^, respectively. The addition of top and bottom surface layers to the gyroid structured specimens significantly increased the bending properties, compressive strength, and SWR. The results of Duncan’s multiple-comparison tests showed that all the treatment groups were significantly different from each other (Table 2). The bending strength and modulus of the specimens improved significantly with the increasing thickness of the face layers. Increasing the surface layer thickness from 0.5 mm to 2.5 mm improved both bending strength and bending modulus from 8.10 N/mm^2^ to 11.82 N/mm^2^ and from 847.5 N/mm^2^ to 2492.2 N/mm^2^, respectively. The rate of increase in the bending modulus (194%) was found to be higher than that for the bending strength (46%) depending on the density of the specimens. In addition, the mass of the specimens increased from 19.4 to 65.8 g depending on the increasing layer thickness, which positively affected the mechanical properties (Table 2). Different failure modes were observed for the bending specimens. The composite panels with a thin layer thickness (0.5 mm and 1 mm) showed ductile fracture behaviour, while the panels with thick surface layers (2 mm and 2.5 mm) failed with a brittle fracture in the core and delamination between surface and core layer after reaching the maximum failure load. The core failure might be explained by the fact that the maximum shear stress reached the critical value (shear strength) of the core material when the thickness of the surface layer was increased.

The bending properties of composite materials are also strongly dependent on the density of the surface layer [23,24,25]. Increasing the ratio of the face and core layers improves the skin of the composites. Since the bending stress is the maximum at the surfaces, an increase in skin density improves the resistance to the load and decreases the deformation of the specimen [26]. This is due to the fact that surface layers have a greater compaction ratio, and consequently better bending properties than the core of the gyroid structure. In particular, compression stress in the top skin layer and tension stress in the bottom skin layer depend on the surface layer thickness and density, which significantly affects the bending properties of the composite materials. This explains why the composite specimens with a higher surface layer ratio had a higher bending strength and bending modulus than the specimens with a lower surface layer ratio. 

The compressive strength (parallel to edgewise) of the rectangular specimens showed a similar trend to the bending properties (Table 2). The compressive strength of the specimens improved with increasing face layer thickness. The compressive strength of the specimens without a face layer was found to be 0.61 N/mm^2^, which was significantly lower than that of the specimens with a face layer. The printing of the surface layer on the gyroid core structure significantly improved the edgewise compressive strength. The compressive strength increased from 3.52 to 14.62 N/mm^2^, a 315% rise, when the surface layer thickness increased from 0.5 mm to 2.5 mm. The failure modes of test specimens after the compressive strength test are presented in Figure 4. Different failure modes were determined depending on the skin thickness of the specimens. As shown in Figure 4, when the surface thickness was as thin as 1 mm delamination was observed between the skin and core. When the skin layer was increased to 2 mm thickness, shear cracks were observed in the core. The maximum load was obtained when the shear cracks propagated vertically in the core of the specimens with 2 and 2.5 mm thick skins. Similar failure modes were observed in previous studies [27,28,29].

The Brinell hardness of the specimens increased significantly with the increasing face thickness of the specimens (Table 2). The load applied by the indenter was 400 N. When the surface thickness was decreased from 2 mm to 1 mm, the Brinell hardness decreased sharply. The Brinell hardness values of the specimens with a thickness of 0.5, 1, 2 and 2.5 mm were found to be 2.12, 3.35, 15.10 and 26 N/mm^2^, respectively. When the face thickness was lower than 2 mm, the deepness and diameter of the indentation increased greatly, and indentation cracks were also observed. However, there were no cracks on the specimens with a face thickness of 2 mm or 2.5 mm (Figure 5). The SWR values increased significantly with increasing surface layer thickness, as expected (Table 2). The SWR is positively affected by the thicker skins because the anchoring depth of the screw is increased. The SWR increased from 445 to 1475 N, a rise of 231%, as the face thickness increased from 0.5 to 2.5 mm. 

## 4. Conclusions

The mechanical properties of 3D printed wood/PLA specimens with a gyroid structure were significantly affected by the face layer thickness. The experimental results revealed that all the mechanical properties improved significantly when the face thickness was increased from 0.5 mm to 2 mm. The results showed that different failure modes were observed for the composites with increasing thickness of the surface layers. The results also show that lightweight structured samples produced from wood/PLA filament by 3D printer can be successfully produced. Based on the findings of the present study, gyroid structured composites with a thickness of 2 mm or more thickness are recommended due to their better mechanical properties as compared to the composites with skins that are thinner.

## Figures and Tables

**Figure 1 polymers-12-02929-f001:**
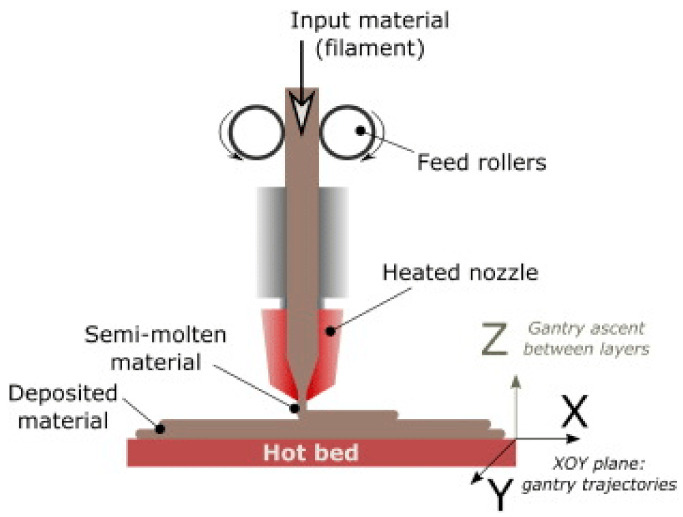
Additive manufacturing (AM) technologies (from [1], with permission).

**Figure 2 polymers-12-02929-f002:**
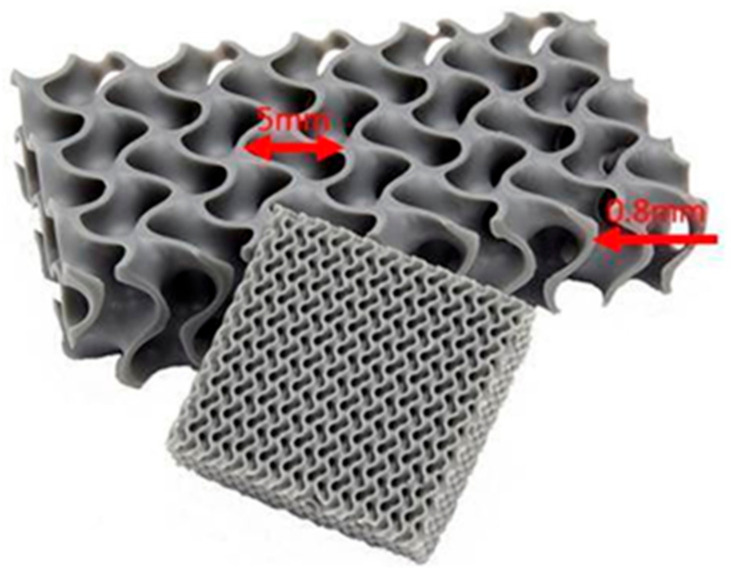
The lattice structures are additively manufactured using wood/PLA filament (reproduced from [17]).

**Figure 3 polymers-12-02929-f003:**
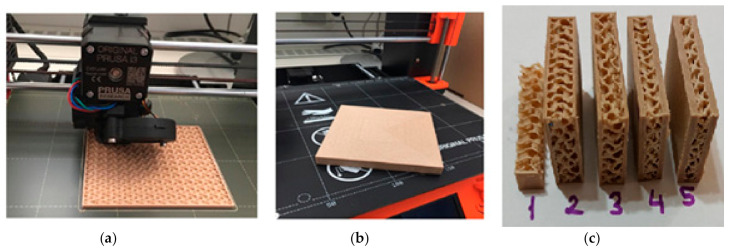
(**a**): The printing of the wood/PLA composite panel in the 3D printer; (**b**): The finished wood/PLA composite panel; (**c**): Cross-section of the specimens with different surface layer thicknesses.

**Figure 4 polymers-12-02929-f004:**
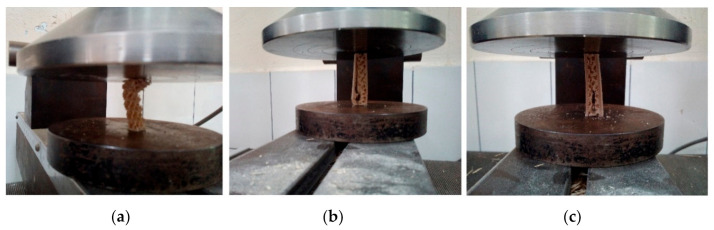
Typical failure modes of compressive test specimens. (**a**) gyroid core without skin; (**b**) Specimens with 1 mm thick skin (vertical failure was between core and skin); (**c**) Specimens with 2 mm thick skin (vertical failure was in the core).

**Figure 5 polymers-12-02929-f005:**
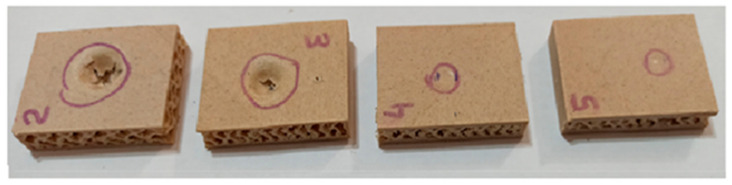
Effect of face thickness on the Brinell hardness of the wood/PLA specimens with a gyroid structure.

**Table 1 polymers-12-02929-t001:** Experimental design of the specimens with a gyroid-structure core.

Specimen Code	Infill Ratio (%)	Sample Thickness (Face/Core/Face) (mm)	Filament Composition	Printing Temperature (°C)	Resolution (mm)	Nozzle Diameter (mm)
1	15	0/10/0	Wood/PLA	210	0.10	0.4
2	15	0.5/9/0.5	Wood/PLA	210	0.10	0.4
3	15	1/8/1	Wood/PLA	210	0.10	0.4
4	15	2/6/2	Wood/PLA	210	0.10	0.4
5	15	2.5/5/2.5	Wood/PLA	210	0.10	0.4

**Table 2 polymers-12-02929-t002:** Mechanical properties of the specimens with a gyroid-structure core.

Specimen Code	Sample Weight ^1^ (g)	Bending Strength (N/mm^2^)	Bending Modulus (N/mm^2^)	Compressive Strength ^2^ (N/mm^2^)	Brinell Hardness (N/mm^2^)	Face Screw Withdrawal Resistance Surface (N)
1	19.4 (0.3) a ^3^	1.76 (0.36) a	324 (5.28) a	0.61 (0.06) a	-	-
2	41.6 (0.4) b	8.10 (0.13) b	847.5 (38) b	3.52 (0.05) b	2.12 (06) a	445 (31) a
3	39.7 (0.4) c	9.74 (0.91) c	1563 (32) c	6.61 (0.33) c	3.35 (0.8) b	635 (39) b
4	56.6 (0.5) d	9.32 (0.82) c	2068.3 (49) d	12.48 (0.51) d	15.10 (1.9) c	1032 (28) c
5	65.8 (0.3) e	11.82 (0.11) d	2492 (41) e	14.62 (0.30) e	26.00 (2.3) d	1475 (54) d

^1^ Sample size: 100 mm × 100 mm × 10 mm. ^2^ Parellel to the surface. deviations. ^3^ Groups with the same letter are in each column not statistically different at a significance level of α = 0.05. The values in the parentheses are standard.

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
