# Peer review of "Effects of the Face/Core Layer Ratio on the Mechanical Properties of 3D Printed Wood/Polylactic Acid (PLA) Green Biocomposite Panels with a Gyroid Core"

_polymers, 2020, doi:10.3390/polym12122929_

Round 1
Reviewer 1 Report
This publication is addressing 3D printing to create PLA Gyroid structures reinforced by wood fiber, which is used as the honeycomb core for sandwich composites.
I am recommending minor revisions, witch will enhance quality of this publication. Below are my comments:
According to my knowledge, 3D printed objects, based on PLA films with wood fibers, are seldom used for structural wood composites, due to its relatively low mechanical properties. Therefore, the best use for this type of composite material would be interior use. However, in the framework of this publication, the mechanical properties are not very important. To show the advantage of these gyroid structures, I would like to suggest (but not impose) that the future study would include one control sample of common honeycomb structure. Such benchmarking should confirm clearly the advantage of those structures. Only then, it would be proven if 3D printed structures are comparable and meaningful.
Another drawback of this paper is that the analytical part is not very strong due to small number of specimens (two specimens per for each treatment type), which makes the statistical analysis less meaningful. In order to convince readers that such low sample size of specimens is adequate, I suggest (but not force) to use t-distribution or other statistical methods. Increasing the number of specimens would enhance the quality of this research but I understand that it may not be always the realistic solution. This paper could be published and open door to future studies.
Article need to be check for small grammatical and punctuation issues, such as those listed below:
Page 1 Line 40: missing period after method.
Page 1 Line 46: missing words – “strength-to-weight ratios of the ???”,
Page 2 Line 48: abrasive the should be “abrasiveness to the”
Page 2 Line 48: “natural sources” should be “natural resources”
Page 2 Line 49: “manu industries” should be “many industries”
Page 2 Line 66: word with 2 x
Page 3 Line 116: Different failure modes or Difference in failure modes
Page 3 Line 124: delete density – repetition
Page 3 Line 130: delete one “two”
Page 3 Line 152: different fond – “well indentation cracks”
Author Response
Dear Many thanks for your valuable comments. Our manuscript has been carefully checked by a Native speaker. We also checked it. We already performed statistical anaylsis on the results.We emphasize the significance of the study in the introduction last paragprah. We explained why this stusy was performed
Thank you
This publication is addressing 3D printing to create PLA Gyroid structures reinforced by wood fiber, which is used as the honeycomb core for sandwich composites. I am recommending minor revisions, witch will enhance quality of this publication. Below are my comments
Response: Thank you for your valuable comments.
Page 1 Line 40: missing period after method.
Page 1 Line 46: missing words – “strength-to-weight ratios of the ???”,
Page 2 Line 48: abrasive the should be “abrasiveness to the”
Page 2 Line 48: “natural sources” should be “natural resources”
Page 2 Line 49: “manu industries” should be “many industries” Page 2 Line 66: word with 2 x / Page 3
Line 116: Different failure modes or Difference in failure modes
Page 3 Line 124: delete density – repetition Page 3 Line 130: delete one “two” Page 3 Line 152: different fond – “well indentation cracks”
Response:All the editorial corrections were performed on the manuscript. The native speaker also checked the manuscript carefully.
Another drawback of this paper is that the analytical part is not very strong due to small number of specimens (two specimens per for each treatment type), which makes the statistical analysis less meaningful. In order to convince readers that such low sample size of specimens is adequate, I suggest (but not force) to use t-distribution or other statistical methods. Increasing the number of specimens would enhance the quality of this research but I understand that it may not be always the realistic solution. This paper could be published and open door to future studies.
Response:In this study, we carried out ANOVA and Duncan’s multipla range test. The test result were statistically comparedfor each treatment group in Table 2. Since the standard deviation of the samples we used ANOVA and Duncan test.

Reviewer 2 Report
This paper prepared a wood/PLA composite by 3D printing and studied the mechanical properties. I am sorry to say that this paper cannot be accepted because it is not a qualified scientific paper, but more like an experimental report.
- There are some format and syntax errors, such as inconsistent font size.
- The introduction is badly organized. Concise expressions and organizational logic are needed to make this work more competitive. What problem do the authors want to solve? Compared to the current work, what are your huge innovations? Please also present the outstanding research challenge that motivates the scope of your study.
- Part 3 “Results and discussion” is just a list of test results, that’s why I say it more like an experimental report. Not only lack of mechanism analysis, but also many important characterization tests, for example, XRD, SEM, and so on.
Author Response
Dear Reviewer
Thank you for your valuable comments. We revised our paper acording to your comments. A native speaker also carefully checked the manuscript.
Thank you
This paper prepared a wood/PLA composite by 3D printing and studied the mechanical properties. I am sorry to say that this paper cannot be accepted because it is not a qualified scientific paper, but more like an experimental report
There are some format and syntax errors, such as inconsistent font size.
Response:All the editorial corrections were performed on the manuscript. The native speaker also checked the manuscript carefully
The introduction is badly organized. Concise expressions and organizational logic are needed to make this work more competitive. What problem do the authors want to solve? Compared to the current work, what are your huge innovations? Please also present the outstanding research challenge that motivates the scope of your study.
Response: We explained the requirement and novelty of this study at the last paragraph of Introduction.
Part 3 “Results and discussion” is just a list of test results, that’s why I say it more like an experimental report. Not only lack of mechanism analysis, but also many important characterization tests, for example, XRD, SEM, and so on.
Response:In the Results section, we discussed the mechanical properties. We used the same filament for all the 3D printed specimens groups. Fort his reason , we did not play with material formulation. The important thing was the effect of skin/core layer ratio, a new core design (Gyroid core structure from wood/PLA flament), and failure modes. Especially, surface hardness, screw withdrawal resistance are significantly affected by skin thickness. It could not find any study on the lightweight composites from wood/PLA filament in the literature.
“Wood has a number of significant advantages when used as a reinforcing filler for thermoplastic composites, such as the low cost, easy supply, high modulus, problem free-disposal, and good machinability. PLA is an environmentally friendly polymer and one of the most used filaments with fused deposition modelling (FDM) printers. The use of wood/PLA composites has increased considerably in the biocomposite industry in recent years, because of their significant advantages as compared to wood-filled petroleum based polymer composites. Wood/PLA composites are mostly produced by extrusion, injection moulding, and hot-press moulding. As compared to the solid wood/PLA composites, lightweight wood/PLA composites are better for using when weight/strength is important. The skin/core layer ratio is one of the most important parameters affecting the mechanical properties of such composites. The ratio of the surface/core layer that can meet the mechanical properties required by the place of use must thus be determined, as otherwise an unsuitable surface/core layer ratio may result in failure under load. A comprehensive search of the literature revealed no studies on the mechanical properties of lightweight composites produced from wood/PLA. The current study thus focused on the impact of the shell ratio on the mechanical properties of the 3D printed lightweight wood/PLA composite panels with a gyroid core structure”.

Round 2
Reviewer 2 Report
The authors have responded well to the previous critiques.